# Multi-Point Boundary Value Problems for $(k, \phi)$-Hilfer Fractional Differential Equations and Inclusions

Jessada Tariboon [1,*,†] , Ayub Samadi [2,*,†] and Sotiris K. Ntouyas [3,†]

1   Intelligent and Nonlinear Dynamic Innovations, Department of Mathematics, Faculty of Applied Science, King Mongkut's University of Technology North Bangkok, Bangkok 10800, Thailand
2   Department of Mathematics, Miyaneh Branch, Islamic Azad University, Miyaneh, Iran
3   Department of Mathematics, University of Ioannina, 451 10 Ioannina, Greece; sntouyas@uoi.gr
*   Correspondence: jessada.t@sci.kmutnb.ac.th (J.T.); ayubtoraj1366@gmail.com (A.S.)
†   These authors contributed equally to this work.

**Abstract:** In this paper we initiate the study of boundary value problems for fractional differential equations and inclusions involving $(k, \phi)$-Hilfer fractional derivative of order in $(1, 2]$. In the single-valued case the existence and uniqueness results are established by using classical fixed-point theorems, such as Banach, Krasnoselskiĭ and Leray-Schauder. In the multivalued case we consider both cases, when the right-hand side has convex or non-convex values. In the first case, we apply the Leray–Schauder nonlinear alternative for multivalued maps, and in the second, the Covit–Nadler fixed-point theorem for multivalued contractions. All results are well illustrated by numerical examples.

**Keywords:** $(k, \phi)$-Hilfer fractional derivative; Riemann-Liouville fractional derivative; Caputo fractional derivative; existence; uniqueness; fixed point theorems

## 1. Introduction and Preliminaries

Fractional calculus and fractional differential equations have cashed substantial consideration owing to the broad applications of fractional derivative operators in the mathematical modelling, describing many real world processes more accurately than the classical-order differential equations. For a systematic development of the topic, see the monographs [1–9]. Fractional derivative operators are usually defined via fractional integral operators. In the literature, many fractional derivative operators have been proposed, such as Riemann–Liouville, Caputo, Hadamard, Erdélyi–Kober and Hilfer fractional operators, to name a few. The Riemann–Liouville fractional integral operator of order $\alpha > 0$ is one of the most used and studied operators, defined by

$$\mathfrak{I}_{a+}^{\alpha}\mathfrak{f}(w) = \frac{1}{\Gamma(\alpha)} \int_a^w (w-u)^{\alpha-1}\mathfrak{f}(u)du, \;\; w > a. \tag{1}$$

The Riemann–Liouvile and Caputo fractional derivative operators of order $\alpha > 0$ are defined in light of the above definition by

$$^{RL}\mathfrak{D}_{a+}^{\alpha}\mathfrak{f}(w) = \mathfrak{D}^{n}\mathfrak{I}_{a+}^{n-\alpha}\mathfrak{f}(w) = \frac{1}{\Gamma(n-\alpha)} \frac{d^n}{dw^n} \int_a^w (w-u)^{n-\alpha-1}\mathfrak{f}(u)du, \;\; w > a, \tag{2}$$

and

$$^{C}\mathfrak{D}_{a+}^{\alpha}\mathfrak{f}(w) = \mathfrak{I}_{a+}^{n-\alpha}\mathfrak{D}^{n}\mathfrak{f}(w) = \frac{1}{\Gamma(n-\alpha)} \int_a^w (w-u)^{n-\alpha-1}\mathfrak{f}^{(n)}(u)du, \;\; w > a, \tag{3}$$

respectively, where $n - 1 < \alpha \le n$ and $n \in \mathbb{N}$. In [10], the Riemann–Liouville fractional integral operator was extended to $k$-Riemann–Liouville fractional integral of order $\alpha > 0$ ($\alpha \in \mathbb{R}$) as

$$^{k}\mathfrak{J}^{\alpha}_{a+}\mathfrak{h}(w) = \frac{1}{k\Gamma_k(\alpha)} \int_a^w (w - u)^{\frac{\alpha}{k}-1} \mathfrak{h}(u) du, \tag{4}$$

where $\mathfrak{h} \in L^1([a,b],\mathbb{R})$, $k > 0$ and $\Gamma_k$ is the $k$-Gamma function for $w \in \mathbb{C}$ with $\Re(w) > 0$ and $k \in \mathbb{R}, k > 0$ which is defined in [11] by

$$\Gamma_k(w) = \int_0^\infty s^{w-1} e^{-\frac{s^k}{k}} ds.$$

The following relations are well known.

$$\Gamma(\theta) = \lim_{k \to 1} \Gamma_k(\theta), \ \Gamma_k(\theta) = k^{\frac{\theta}{k}-1}\Gamma\left(\frac{\theta}{k}\right) \text{ and } \Gamma_k(\theta + k) = \theta\Gamma_k(\theta).$$

In [12] the $k$-Riemann–Liouville fractional derivative was introduced as

$$^{k,RL}\mathfrak{D}^{\alpha}_{a+}\mathfrak{h}(w) = \left(k\frac{d}{dw}\right)^n {}^{k}\mathfrak{J}^{nk-\alpha}_{a+}\mathfrak{h}(w), \ \ n = \left\lceil\frac{\alpha}{k}\right\rceil, \tag{5}$$

where $\mathfrak{h} \in L^1([a,b],\mathbb{R})$, $k, \alpha \in \mathbb{R}^+$ and $\left\lceil\frac{\alpha}{k}\right\rceil$ is the ceiling function of $\frac{\alpha}{k}$.

On the other hand in [2] the $\phi$-Riemann–Liouville fractional integral of the function $\mathfrak{h} \in L^1([a,b],\mathbb{R})$ and an increasing function $\phi : [a,b] \to \mathbb{R}$ with $\phi'(w) \ne 0$ for all $w \in [a,b]$, was given by

$$\mathfrak{J}^{\bar{\alpha};\phi}\mathfrak{h}(w) = \frac{1}{\Gamma_k(\alpha)} \int_a^w \phi'(u)(\phi(w) - \phi(u))^{\bar{\alpha}-1}\mathfrak{h}(u) du. \tag{6}$$

Let $n - 1 < \bar{\alpha} \le n$, $\phi \in C^n([a,b],\mathbb{R})$, $\phi'(w) \ne 0, w \in [a,b]$, and $\mathfrak{h} \in C([a,b],\mathbb{R})$. Then the $\phi$-Riemann–Liouville fractional derivative of the function $\mathfrak{h}$ of order $\bar{\alpha}$ was defined in [2] by

$$^{RL}\mathfrak{D}^{\bar{\alpha};\phi}\mathfrak{h}(w) = \left(\frac{1}{\phi'(w)}\frac{d}{dw}\right)^n \mathfrak{J}^{n-\bar{\alpha};\phi}_{a+}\mathfrak{h}(w), \tag{7}$$

and the $\phi$-Caputo fractional derivative of the function $\mathfrak{h}$ of order $\alpha$ was defined in [13] by

$$^{C}\mathfrak{D}^{\bar{\alpha};\phi}\mathfrak{h}(w) = \mathfrak{J}^{n-\bar{\alpha};\phi}_{a+}\left(\frac{1}{\phi'(w)}\frac{d}{dw}\right)^n \mathfrak{h}(w), \tag{8}$$

respectively. In [14] the $\phi$-Hilfer fractional derivative of the function $\mathfrak{h} \in C([a,b],\mathbb{R})$ of order $\bar{\alpha} \in (n-1,n]$ and type $\beta \in [0,1]$ and $\phi \in C^n([a,b],\mathbb{R})$, $\phi'(w) \ne 0, w \in [a,b]$, was defined by

$$^{H}\mathfrak{D}^{\bar{\alpha},\beta;\phi}\mathfrak{h}(w) = \mathfrak{J}^{\beta(n-\bar{\alpha});\phi}_{a+}\left(\frac{1}{\phi'(w)}\frac{d}{dw}\right)^n \mathfrak{J}^{(1-\beta)(n-\bar{\alpha});\phi}_{a+}\mathfrak{h}(w). \tag{9}$$

In [15] was defined the $(k,\phi)$-Riemann–Liouville fractional integral of order $\bar{\alpha} > 0$ ($\alpha \in \mathbb{R}$) of the function $\mathfrak{h} \in L^1([a,b],\mathbb{R})$, $k > 0$, as

$$^{k}\mathfrak{J}^{\bar{\alpha};\phi}_{a+}\mathfrak{h}(w) = \frac{1}{k\Gamma_k(\bar{\alpha})} \int_a^w \phi'(u)(\phi(w) - \phi(u))^{\frac{\bar{\alpha}}{k}-1}\mathfrak{h}(u) du. \tag{10}$$

Recently, in [16] introduced $(k,\phi)$-Hilfer fractional derivative of the function $\mathfrak{h} \in C^n([a,b],\mathbb{R})$ of order $\bar{\alpha} > 0, k > 0$ and type $\beta \in [0,1]$, $\phi \in C^n([a,b],\mathbb{R})$, $\phi'(w) \ne 0, w \in [a,b]$ as

$$^{k,H}\mathfrak{D}^{\bar{\alpha},\beta;\phi}\mathfrak{h}(w) = {}^{k}\mathfrak{J}^{\beta(nk-\bar{\alpha});\phi}_{a+}\left(\frac{k}{\phi'(w)}\frac{d}{dw}\right)^n {}^{k}\mathfrak{J}^{(1-\beta)(nk-\bar{\alpha});\phi}_{a+}\mathfrak{h}(w), \ \ n = \left\lceil\frac{\bar{\alpha}}{k}\right\rceil. \tag{11}$$

Note that:

1.  For $\beta = 0$, (11) reduces to $(k, \phi)$-Riemann–Liouville fractional derivative operator

    $$^{k,RL}\mathfrak{D}^{\bar{\alpha};\phi}\mathfrak{h}(w) = \left( \frac{k}{\phi'(w)} \frac{d}{dw} \right)^n {}^k\mathfrak{I}_{a+}^{(1-\beta)(nk-\bar{\alpha});\phi}\mathfrak{h}(w). \tag{12}$$

    If we take in (12), $\phi(w) = w$, then we obtain $k$-Riemann–Liouville fractional derivative operator defined in [12];

2.  For $\beta = 1$, (11) reduces to $(k, \phi)$-Caputo fractional derivative operator [16]

    $$^{k,C}\mathfrak{D}^{\bar{\alpha};\phi}\mathfrak{h}(w) = {}^k\mathfrak{I}_{a+}^{nk-\bar{\alpha};\phi}\left( \frac{k}{\phi'(w)} \frac{d}{dw} \right)^n \mathfrak{h}(w). \tag{13}$$

    If we take $\phi(w) = w$ in (13), then we obtain $k$-Caputo fractional derivative operator [16]

    $$^{k,C}\mathfrak{D}^{\bar{\alpha};\phi}\mathfrak{h}(w) = {}^k\mathfrak{I}_{a+}^{nk-\bar{\alpha};\phi}\left( k\frac{d}{dw} \right)^n \mathfrak{h}(w). \tag{14}$$

3.  If $\phi(w) = w^\rho$, then (11) reduces to $k$-Hilfer–Katugampola fractional derivative operator:
    (a)  If $\phi(w) = w^\rho$, $\beta = 0$, then (11) reduces to $k$-Katugampola fractional derivative operator [17];
    (b)  If $\phi(w) = w^\rho$, $\beta = 1$, then (11) reduces to $k$-Caputo–Katugampola fractional derivative operator [17];

4.  If $\phi(w) = \log w$, then (11) reduces to $k$-Hilfer–Hadamard fractional derivative operator:
    (a)  If $\phi(w) = \log w$, $\beta = 0$, then (11) reduces to $k$-Hadamard fractional derivative operator [16];
    (b)  If $\phi(w) = \log w$, $\beta = 1$, then (11) reduces to $k$-Caputo–Hadamard fractional derivative operator [16].

**Remark 1.** *If $\theta_k = \bar{\alpha} + \beta(nk - \bar{\alpha})$, then $\beta(nk - \bar{\alpha}) = \theta_k - \bar{\alpha}$ and $(1 - \beta)(nk - \bar{\alpha}) = nk - \theta_k$ and hence the $(k, \phi)$-Hilfer fractional derivative has been defined in the form of $(k, \phi)$-Riemann-Liouville fractional derivative as follows*

$$\begin{aligned}
^{k,H}\mathfrak{D}^{\bar{\alpha},\beta;\phi}\mathfrak{h}(w) &= {}^k\mathfrak{I}_{a+}^{\theta_k-\bar{\alpha};\phi}\left( \frac{k}{\phi'(w)} \frac{d}{dw} \right)^n {}^k\mathfrak{I}_{a+}^{nk-\theta_k;\phi}\mathfrak{h}(w) \\
&= {}^k\mathfrak{I}_{a+}^{\theta_k-\bar{\alpha};\phi}\left( {}^{k,RL}\mathfrak{D}^{\theta_k;\phi}\mathfrak{h} \right)(w).
\end{aligned}$$

*Note for $\beta \in [0, 1]$ and $n - 1 < \frac{\bar{\alpha}}{k} \le n$, we have $n - 1 < \frac{\theta_k}{k} \le n$.*

For some results on $k$-Riemann–Liouville fractional derivatives, we refer to [18–23] and the therein-cited references.

In [16] the authors proved several properties of $(k, \phi)$-Hilfer fractional derivative operator. Moreover they studied the following nonlinear initial value problem involving $(k, \phi)$-Hilfer fractional derivative of the form

$$\begin{cases} ^{k,H}\mathfrak{D}_{a+}^{\bar{\alpha},\beta;\phi}\vartheta(w) = \mathfrak{f}(w, \vartheta(w)), \ w \in (a, b], \ 0 < \bar{\alpha} < k, \ 0 \le \beta \le 1, \\ {}^k\mathfrak{I}^{k-\theta_k;\phi}\vartheta(a) = x_a \in \mathbb{R}, \ \theta_k = \bar{\alpha} + \beta(k - \bar{\alpha}), \end{cases} \tag{15}$$

where $^{k,H}\mathfrak{D}^{\bar{\alpha},\beta;\phi}$ denotes the $(k, \phi)$-Hilfer fractional derivative operator of order $\bar{\alpha}, 0 < \bar{\alpha} \le 1$ and parameter $\beta, 0 \le \beta \le 1$, and $\mathfrak{f} : [a, b] \times \mathbb{R} \to \mathbb{R}$ is a continuous function. By applying Banach's fixed point theorem they proved the existence of a unique solution for the problem (15).

In the present work, motivated by the paper [16], we study boundary value problems involving $(k, \phi)$-Hilfer fractional derivative operator of order $\bar{\alpha}$ and parameter $\beta$, where $1 < \bar{\alpha} \le 2$ and $0 \le \beta \le 1$. To be more precisely, we consider in this paper the following $(k, \phi)$-

Hilfer fractional boundary value problem with nonlocal multipoint boundary conditions of the form

$$
\begin{cases}
{}^{k,H}\mathfrak{D}^{\bar{\alpha},\beta;\phi}\vartheta(w) = \mathfrak{f}(w,\vartheta(w)), & w \in (a,b], \\
\vartheta(a) = 0, & \vartheta(b) = \displaystyle\sum_{i=1}^{m} \lambda_i \vartheta(\xi_i),
\end{cases}
\tag{16}
$$

where ${}^{k,H}\mathfrak{D}^{\bar{\alpha},\beta;\phi}$ denotes the $(k,\phi)$-Hilfer fractional derivative operator of order $\bar{\alpha}$, $1 < \bar{\alpha} < 2$ and parameter $\beta$, $0 \le \beta \le 1$, $k > 0$, $\mathfrak{f} : [a,b] \times \mathbb{R} \to \mathbb{R}$ is a continuous function, $\lambda_i \in \mathbb{R}$, and $a < \xi_i < b, i = 1, 2, \ldots, m$. Our aim in this paper is to establish results concerning existence and uniqueness, by using Banach's and Krasnoselskiĭ's fixed point theorems, as well as a Leray–Schauder nonlinear alternative.

Next, we also study the multivalued problem

$$
\begin{cases}
{}^{k,H}\mathfrak{D}^{\bar{\alpha},\beta;\phi}\vartheta(w) \in \mathfrak{F}(w,\vartheta(w)), & w \in (a,b], \\
\vartheta(a) = 0, & \vartheta(b) = \displaystyle\sum_{i=1}^{m} \lambda_i \vartheta(\xi_i),
\end{cases}
\tag{17}
$$

in which $\mathfrak{F} : [a,b] \times \mathbb{R} \to \mathcal{P}(\mathbb{R})$ is a multivalued map and the other parameters are as in problem (16). Here, $\mathcal{P}(\mathbb{R})$ denotes the family of all nonempty subsets of $\mathbb{R}$. We will study both cases, when the right-hand side is convex or nonconvex valued, and we will establish existence results by using Leray–Schauder nonlinear alternative for multivalued maps and the Covitz–Nadler fixed-point theorem for multivalued contractions, respectively.

Numerical examples are constructed illustrating the applicability of our obtained theoretical results.

The rest of our paper is organized as follows. In Section 2, we prove an ancillary result toward a linear variant of the $(k,\phi)$-Hilfer fractional nonlocal boundary value problem (16). This lemma is important to transform the nonlinear boundary value problem (16) into an equivalent fixed-point problem. The main results for the single valued $(k,\phi)$-Hilfer fractional nonlocal boundary value problem (16) are included in Section 3, while the results for the multivalued $(k,\phi)$-Hilfer fractional nonlocal boundary value problem (17) are presented in Section 4. Finally, Section 5 is dedicated to illustrative examples.

## 2. An Auxiliary Result

In this section an auxiliary result is proved, which is the basic tool in transforming the nonlinear problem (16) into a fixed-point problem, and dealing with a linear variant of the problem (16). First we recall two useful lemmas.

**Lemma 1** ([16]). *Let* $\mu, k \in \mathbb{R}^+ = (0,\infty)$ *and* $n = \left\lceil \frac{\mu}{k} \right\rceil$. *Assume that* $\mathfrak{h} \in C^n([a,b],\mathbb{R})$ *and* ${}^{k}\mathfrak{J}_{a+}^{nk-\mu;\phi}\mathfrak{h} \in C^n([a,b],\mathbb{R})$. *Then*

$$
{}^{k}\mathfrak{J}^{\mu;\phi}\left({}^{k,RL}\mathfrak{D}^{\mu;\phi}\mathfrak{h}(w)\right) = \mathfrak{h}(w) - \sum_{j=1}^{n} \frac{(\phi(w)-\phi(a))^{\frac{\mu}{k}-j}}{\Gamma_k(\mu - jk + k)} \left[ \left( \frac{k}{\phi'(w)} \frac{d}{dw} \right)^{n-j} {}^{k}\mathfrak{J}_{a+}^{nk-\mu;\phi}\mathfrak{h}(w) \right]_{z=a}.
$$

**Lemma 2** ([16]). *Let* $\alpha, k \in \mathbb{R}^+ = (0,\infty)$ *with* $\alpha < k$, $\beta \in [0,1]$ *and* $\theta_k = \alpha + \beta(k - \alpha)$. *Then*

$$
{}^{k}\mathfrak{J}^{\theta_k;\phi}\left({}^{k,RL}\mathfrak{D}^{\theta_k;\phi}\mathfrak{h}\right)(w) = {}^{k}\mathfrak{J}^{\alpha;\phi}\left({}^{k,H}\mathfrak{D}^{\alpha,\beta;\phi}\mathfrak{h}\right)(w), \quad \mathfrak{h} \in C^n([a,b],\mathbb{R}).
$$

**Lemma 3.** *Let* $a < b, k > 0, 1 < \bar{\alpha} \le 2, \beta \in [0,1], \theta_k = \bar{\alpha} + \beta(2k - \bar{\alpha}), \mathfrak{g} \in C^2([a,b],\mathbb{R})$ *and*

$$
\mathcal{H} := \frac{1}{\Gamma_k(\theta_k)} \left[ (\phi(b)-\phi(a))^{\frac{\theta_k}{k}-1} - \sum_{i=1}^{m} \lambda_i (\phi(\xi_i)-\phi(a))^{\frac{\theta_k}{k}-1} \right] \neq 0.
\tag{18}
$$

*Then the function $\vartheta \in C([a, b], \mathbb{R})$ is a solution of the boundary value problem*

$$\begin{cases} {}^{k,H}\mathfrak{D}^{\bar{\alpha},\beta;\phi}\vartheta(w) = \mathfrak{g}(w), & w \in (a, b], \\ \vartheta(a) = 0, & \vartheta(b) = \sum_{i=1}^{m} \lambda_i \vartheta(\xi_i), \end{cases} \tag{19}$$

*if and only if*

$$\vartheta(w) = {}^{k}\mathfrak{I}^{\bar{\alpha};\phi}\mathfrak{g}(w) + \frac{(\phi(w) - \phi(a))^{\frac{\theta_k}{k}-1}}{\mathcal{H}\Gamma_k(\theta_k)}\left[\sum_{i=1}^{m}\lambda_i\,{}^{k}\mathfrak{I}^{\bar{\alpha};\phi}\mathfrak{g}(\xi_i) - {}^{k}\mathfrak{I}^{\bar{\alpha};\phi}\mathfrak{g}(b)\right]. \tag{20}$$

**Proof.** Assume that $\vartheta$ is a solution of the boundary value problem (19). Operating fractional integral ${}^{k}\mathfrak{I}^{\alpha;\phi}$ on both sides of equation in (19) and using Lemmas 1 and 2, we obtain

$$\begin{aligned}{}^{k}\mathfrak{I}^{\bar{\alpha};\phi}\left({}^{k,H}\mathfrak{D}^{\bar{\alpha},\beta;\phi}\vartheta\right)(w) &= {}^{k}\mathfrak{I}^{\theta_k;\phi}\left({}^{k,RL}\mathfrak{D}^{\theta_k;\phi}\vartheta\right)(w) \\ &= \vartheta(w) - \frac{(\phi(w) - \phi(a))^{\frac{\theta_k}{k}-1}}{\Gamma_k(\theta_k)}\left[\left(\frac{k}{\phi'(w)}\frac{d}{dw}\right){}^{k}\mathfrak{I}^{2k-\theta_k;\phi}\vartheta(w)\right]_{w=a} \\ &\quad - \frac{(\phi(w) - \phi(a))^{\frac{\theta_k}{k}-2}}{\Gamma_k(\theta_k - k)}\left[{}^{k}\mathfrak{I}^{2k-\theta_k;\phi}\vartheta(w)\right]_{w=a}.\end{aligned}$$

Consequently

$$\vartheta(w) = {}^{k}\mathfrak{I}^{\bar{\alpha};\phi}\mathfrak{g}(w) + c_0\frac{(\phi(w) - \phi(a))^{\frac{\theta_k}{k}-1}}{\Gamma_k(\theta_k)} + c_1\frac{(\phi(w) - \phi(a))^{\frac{\theta_k}{k}-2}}{\Gamma_k(\theta_k - k)}, \tag{21}$$

where

$$c_0 = \left[\left(\frac{k}{\phi'(w)}\frac{d}{dw}\right){}^{k}\mathfrak{I}^{2k-\theta_k;\phi}\vartheta(w)\right]_{w=a}, \quad c_1 = \left[{}^{k}\mathfrak{I}^{2k-\theta_k;\phi}\vartheta(w)\right]_{w=a}.$$

From the boundary condition $\vartheta(a) = 0$ we get $c_2 = 0$, since $\frac{\theta_k}{k} - 2 < 0$ by Remark 1. From the second boundary condition $\vartheta(b) = \sum_{i=1}^{m}\lambda_i\vartheta(\xi_i)$ we found

$$c_0 = \frac{1}{\mathcal{H}}\left[\sum_{i=1}^{m}\lambda_i\,{}^{k}\mathfrak{I}^{\bar{\alpha};\phi}\mathfrak{g}(\xi_i) - {}^{k}\mathfrak{I}^{\bar{\alpha};\phi}\mathfrak{g}(b)\right].$$

Replacing the values of $c_0$ and $c_1$ in (21), we get the solution (20). We can prove easily the converse by direct computation. The proof is finished. □

## 3. The Single Valued Problem

Let $C([a, b], \mathbb{R})$ be the Banach space of all continuous functions from $[a, b]$ to $\mathbb{R}$ endowed with the sup-norm $\|\vartheta\| = \sup_{w\in[a,b]}|\vartheta(w)|$. In view of Lemma 3, we define an operator $\mathcal{A} : C([a, b], \mathbb{R}) \to C([a, b], \mathbb{R})$ by

$$\begin{aligned}(\mathcal{A}\vartheta)(w) &= \frac{(\phi(w) - \phi(a))^{\frac{\theta_k}{k}-1}}{\mathcal{H}\Gamma_k(\theta_k)}\left[\sum_{i=1}^{m}\lambda_i\,{}^{k}\mathfrak{I}^{\bar{\alpha};\phi}\mathfrak{f}(\xi_i, \vartheta(\xi_i)) - {}^{k}\mathfrak{I}^{\bar{\alpha};\phi}\mathfrak{f}(b, \vartheta(b))\right] \\ &\quad + {}^{k}\mathfrak{I}^{\bar{\alpha};\phi}\mathfrak{f}(w, \vartheta(w)), \quad w \in [a, b].\end{aligned} \tag{22}$$

It should be noticed that the solutions of the nonlocal $(k, \phi)$-Hilfer fractional boundary value problem (16) will be fixed points of $\mathcal{A}$.

For convenience we put:

$$
\begin{aligned}
\mathfrak{G} \;=\;& \frac{(\phi(b)-\phi(a))^{\frac{\bar{\alpha}}{k}}}{\Gamma_k(\bar{\alpha}+k)} + \frac{(\phi(b)-\phi(a))^{\frac{\theta_k}{k}-1}}{|\mathcal{H}|\Gamma_k(\theta_k)} \Bigg[ \sum_{i=1}^{m} |\lambda_i| \frac{(\phi(\xi_i)-\phi(a))^{\frac{\bar{\alpha}}{k}}}{\Gamma_k(\bar{\alpha}+k)} \\
& + \frac{(\phi(b)-\phi(a))^{\frac{\bar{\alpha}}{k}}}{\Gamma_k(\bar{\alpha}+k)} \Bigg].
\end{aligned} \tag{23}
$$

### 3.1. Existence of a Unique Solution

In our first result we will prove the existence of a unique solution of the problem (16). The basic tool is the Banach's contraction mapping principle [24].

**Theorem 1.** *Assume that:*

$(H_1)$ $|\mathfrak{f}(w,\vartheta) - \mathfrak{f}(w,y)| \leq \mathfrak{L}|\vartheta - y|$, $\mathfrak{L} > 0$ *for each* $w \in [a,b]$ *and* $\vartheta, y \in \mathbb{R}$.

*Then the $(k,\phi)$-Hilfer nonlocal multi-point fractional boundary value problem (16) has a unique solution on $[a,b]$, provided that*

$$
\mathfrak{L}\mathfrak{G} < 1, \tag{24}
$$

*where $\mathfrak{G}$ is defined by (23).*

**Proof.** We transform the $(k,\phi)$-Hilfer nonlocal multipoint fractional boundary value problem (16) into a fixed-point problem, with the help of the operator $\mathcal{A}$ defined in (22). Then, we shall show that the operator $\mathcal{A}$ has a unique fixed point.

We let $\sup_{w\in[a,b]} |\mathfrak{f}(w,0)| = \mathfrak{M} < \infty$, and choose

$$
r \geq \frac{\mathfrak{M}\mathfrak{G}}{1 - \mathfrak{L}\mathfrak{G}}. \tag{25}
$$

Let $B_r = \{\vartheta \in C([a,b],\mathbb{R}) : \|\vartheta\| \leq r\}$. In the first step we will show that $\mathcal{A}B_r \subset B_r$. We have, for $\vartheta \in B_r$, using $(H_1)$, that

$$
\begin{aligned}
|\mathfrak{f}(w,\vartheta(w))| &\leq |\mathfrak{f}(w,\vartheta(w)) - \mathfrak{f}(w,0)| + |\mathfrak{f}(w,0)| \\
&\leq \mathfrak{L}|\vartheta(w)| + \mathfrak{M} \leq \mathfrak{L}\|\vartheta\| + \mathfrak{M} \leq \mathfrak{L}r + \mathfrak{M}.
\end{aligned}
$$

For any $\vartheta \in B_r$, we have

$$
\begin{aligned}
|(\mathcal{A}\vartheta)(w)| \;\leq\;& \sup_{w\in[a,b]} \Bigg\{ \frac{(\phi(w)-\phi(a))^{\frac{\theta_k}{k}-1}}{|\mathcal{H}|\Gamma_k(\theta_k)} \Big[ \sum_{i=1}^{m} |\lambda_i|{}^k\mathfrak{J}^{\bar{\alpha};\phi}|\mathfrak{f}(\xi_i,\vartheta(\xi_i))| + {}^k\mathfrak{J}^{\bar{\alpha};\phi}|\mathfrak{f}(b,\vartheta(b))| \Big] \\
& + {}^k\mathfrak{J}^{\bar{\alpha};\phi}|\mathfrak{f}(w,\vartheta(w))| \Bigg\} \\
\leq\;& {}^k\mathfrak{J}^{\bar{\alpha};\phi}\big(|\mathfrak{f}(w,\vartheta(w)) - \mathfrak{f}(w,0)| + |\mathfrak{f}(w,0)|\big) \\
& + \frac{(\phi(b)-\phi(a))^{\frac{\theta_k}{k}-1}}{|\mathcal{H}|\Gamma_k(\theta_k)} \Big( \sum_{i=1}^{m} |\lambda_i|{}^k\mathfrak{J}^{\bar{\alpha};\phi}|\mathfrak{f}(\xi_i,\vartheta(\xi_i)) - \mathfrak{f}(\xi_i,0)| + |\mathfrak{f}(\xi_i,0)| \Big) \\
& + {}^k\mathfrak{J}^{\bar{\alpha};\phi}\big(|\mathfrak{f}(b,\vartheta(b)) - \mathfrak{f}(b,0)| + |\mathfrak{f}(b,0)|\big) \Big) \\
\leq\;& \Bigg\{ \frac{(\phi(b)-\phi(a))^{\frac{\bar{\alpha}}{k}}}{\Gamma_k(\alpha+k)} + \frac{(\phi(b)-\phi(a))^{\frac{\theta_k}{k}-1}}{|\mathcal{H}|\Gamma_k(\theta_k)} \Bigg[ \sum_{i=1}^{m} |\lambda_i| \frac{(\phi(\xi_i)-\phi(a))^{\frac{\bar{\alpha}}{k}}}{\Gamma_k(\alpha+k)} \\
& + \frac{(\phi(b)-\phi(a))^{\frac{\bar{\alpha}}{k}}}{\Gamma_k(\alpha+k)} \Bigg] \Bigg\} (\mathfrak{L}\|\vartheta\| + \mathfrak{M}) \\
\leq\;& (\mathfrak{L}r + \mathfrak{M})\mathfrak{G} \leq r.
\end{aligned}
$$

Consequently $\|\mathcal{A}\vartheta\| \leq r$ and thus $\mathcal{A}B_r \subset B_r$.

Now we will show that $\mathcal{A}$ is a contraction. For $w \in [a, b]$ and $\vartheta, y \in C([a,b], \mathbb{R})$, we have

$$
\begin{aligned}
&|(\mathcal{A}\vartheta)(w) - (\mathcal{A}y)(w)| \\
\leq\quad & {}^{k}\mathfrak{J}^{\overline{\alpha};\phi}|\mathfrak{f}(w, \vartheta(w)) - \mathfrak{f}(w, y(w))| \\
& + \frac{(\phi(b) - \phi(a))^{\frac{\theta_k}{k} - 1}}{|\mathcal{H}|\Gamma_k(\theta_k)}\Big( \sum_{i=1}^{m} |\lambda_i|\,{}^{k}\mathfrak{J}^{\overline{\alpha};\phi}|\mathfrak{f}(\xi_i, \vartheta(\xi_i)) - \mathfrak{f}(\xi_i, y(\xi_i))| \\
& + {}^{k}\mathfrak{J}^{\overline{\alpha};\phi}\big(|\mathfrak{f}(b, \vartheta(b)) - \mathfrak{f}(b, y(b))|\big)\Big) \\
\leq\quad & \Bigg\{ \frac{(\phi(b) - \phi(a))^{\frac{\overline{\alpha}}{k}}}{\Gamma_k(\overline{\alpha} + k)} + \frac{(\phi(b) - \phi(a))^{\frac{\theta_k}{k} - 1}}{|\mathcal{H}|\Gamma_k(\theta_k)}\Bigg[ \sum_{i=1}^{m} |\lambda_i|\frac{(\phi(\xi_i) - \phi(a))^{\frac{\overline{\alpha}}{k}}}{\Gamma_k(\overline{\alpha} + k)} \\
& + \frac{(\phi(b) - \phi(a))^{\frac{\overline{\alpha}}{k}}}{\Gamma_k(\overline{\alpha} + k)}\Bigg] \Bigg\} \mathfrak{L}\|x - y\| \\
=\quad & \mathfrak{L}\mathfrak{G}\|x - y\|.
\end{aligned}
$$

Hence $\|\mathcal{A}x - \mathcal{A}y\| \leq \mathfrak{L}\mathfrak{G}\|x - y\|$ which implies that $\mathcal{A}$ is a contraction, since $\mathfrak{L}\mathfrak{G} < 1$. By the Banach's contraction-mapping principle, the operator $\mathcal{A}$ has a unique fixed point, which is the unique solution of $(k, \phi)$-Hilfer nonlocal multipoint fractional boundary value problem (16). The proof is finished. $\quad\square$

### 3.2. Existence Results

In the forthcoming theorems we will prove existence results for the $(k, \phi)$-Hilfer nonlocal multipoint fractional boundary value problem (16), utilizing Krasnoselskiĭ's fixed point theorem [25] and nonlinear alternative of Leray–Schauder type [26].

**Theorem 2.** *Let* $\mathfrak{f} : [a, b] \times \mathbb{R} \to \mathbb{R}$ *be a continuous function satisfying* $(H_1)$. *In addition we assume that:*

$(H_2)$ $|\mathfrak{f}(w, \vartheta)| \leq \varpi(w), \quad \forall (w, \vartheta) \in [a, b] \times \mathbb{R}, and \ \varpi \in C([a, b], \mathbb{R}^+).$

*Then the* $(k, \phi)$-*Hilfer nonlocal multi-point fractional boundary value problem (16) has at least one solution on* $[a, b]$, *if* $\mathfrak{G}_1\mathfrak{L} < 1$, *where*

$$
\mathfrak{G}_1 := \frac{(\phi(b) - \phi(a))^{\frac{\theta_k}{k} - 1}}{|\mathcal{H}|\Gamma_k(\theta_k)}\left[ \sum_{i=1}^{m} |\lambda_i|\frac{(\phi(\xi_i) - \phi(a))^{\frac{\overline{\alpha}}{k}}}{\Gamma_k(\overline{\alpha} + k)} + \frac{(\phi(b) - \phi(a))^{\frac{\overline{\alpha}}{k}}}{\Gamma_k(\overline{\alpha} + k)} \right]. \tag{26}
$$

**Proof.** Set $\sup_{w \in [a,b]} \varpi(w) = \|\varpi\|$ and $B_\rho = \{\vartheta \in C([a,b], \mathbb{R}) : \|\vartheta\| \leq \rho\}$, with $\rho \geq \|\varpi\|\mathfrak{G}$. We define on $B_\rho$ two operators $\mathcal{A}_1, \mathcal{A}_2$ by

$$
\mathcal{A}_1\vartheta(w) = {}^{k}\mathfrak{J}^{\overline{\alpha};\phi}\mathfrak{f}(w, \vartheta(w)), \quad w \in [a, b],
$$

and

$$
\mathcal{A}_2\vartheta(w) = \frac{(\phi(w) - \phi(a))^{\frac{\theta_k}{k} - 1}}{\mathcal{H}\Gamma_k(\theta_k)}\left[ \sum_{i=1}^{m} \lambda_i\,{}^{k}\mathfrak{J}^{\overline{\alpha};\phi}\mathfrak{f}(\xi_i, \vartheta(\xi_i)) - {}^{k}\mathfrak{J}^{\overline{\alpha};\phi}\mathfrak{f}(b, \vartheta(b)) \right], \quad w \in [a, b].
$$

For any $\vartheta, y \in B_\rho$, we have

$$|(\mathcal{A}_1\vartheta)(w) + (\mathcal{A}_2 y)(w)|$$

$$\leq \sup_{w\in[a,b]} \left\{ \frac{(\phi(w)-\phi(a))^{\frac{\theta_k}{k}-1}}{|\mathcal{H}|\Gamma_k(\theta_k)} \left[ \sum_{i=1}^m |\lambda_i| {}^k\mathfrak{I}^{\overline{\alpha};\phi}|\mathfrak{f}(\xi_i, y(\xi_i))| + {}^k\mathfrak{I}^{\overline{\alpha};\phi}|\mathfrak{f}(b, y(b))| \right] \right.$$

$$\left. + {}^k\mathfrak{I}^{\overline{\alpha};\phi}|\mathfrak{f}(w, \vartheta(w))| \right\}$$

$$\leq \left\{ \frac{(\phi(b)-\phi(a))^{\frac{\alpha}{k}}}{\Gamma_k(\overline{\alpha}+k)} + \frac{(\phi(b)-\phi(a))^{\frac{\theta_k}{k}-1}}{|\mathcal{H}|\Gamma_k(\theta_k)} \left[ \sum_{i=1}^m |\lambda_i| \frac{(\phi(\xi_i)-\phi(a))^{\frac{\overline{\alpha}}{k}}}{\Gamma_k(\overline{\alpha}+k)} \right. \right.$$

$$\left. \left. + \frac{(\phi(b)-\phi(a))^{\frac{\overline{\alpha}}{k}}}{\Gamma_k(\overline{\alpha}+k)} \right] \right\} \|\varpi\|$$

$$= \mathfrak{G}\|\varpi\| \leq \rho.$$

Therefore $\|(\mathcal{A}_1\vartheta) + (\mathcal{A}_2 y)\| \leq \rho$, which shows that $\mathcal{A}_1\vartheta + \mathcal{A}_2 y \in B_\rho$. Next we show that $\mathcal{A}_2$ is a contraction mapping. We omit the details since it is easy by using (26).

The operator $\mathcal{A}_1$ is continuous, since $\mathfrak{f}$ is continuous. Moreover, $\mathcal{A}_1$ is uniformly bounded on $B_\rho$ as

$$\|\mathcal{A}_1\vartheta\| \leq \frac{(\phi(b)-\phi(a))^{\frac{\overline{\alpha}}{k}}}{\Gamma_k(\overline{\alpha}+k)} \|\varpi\|.$$

To prove the compactness of the operator $\mathcal{A}_1$, we consider $w_1, w_2 \in [a,b]$ with $w_1 < w_2$. Then we have

$$|(\mathcal{A}_1\vartheta)(w_2) - (\mathcal{A}_1\vartheta)(w_1)|$$

$$\leq \frac{1}{\Gamma_k(\overline{\alpha})} \left| \int_a^{w_1} \phi'(s)[(\phi(w_2)-\phi(s))^{\frac{\overline{\alpha}}{k}-1} - (\phi(w_1)-\phi(s))^{\frac{\overline{\alpha}}{k}-1}]\mathfrak{f}(s, \vartheta(s))ds \right.$$

$$\left. + \int_{w_1}^{w_2} \phi'(s)(\phi(w_2)-\phi(s))^{\frac{\overline{\alpha}}{k}-1}\mathfrak{f}(s, \vartheta(s))ds \right|$$

$$\leq \frac{\|\varpi\|}{\Gamma_k(\overline{\alpha}+k)} [2(\phi(w_2)-\phi(w_1))^{\frac{\overline{\alpha}}{k}} + |(\phi(w_2)-\phi(a))^{\frac{\overline{\alpha}}{k}} - (\phi(w_1)-\phi(a))^{\frac{\overline{\alpha}}{k}}|],$$

which tends to zero as $w_2 - w_1 \to 0$, independently of $\vartheta$. Thus, $\mathcal{A}_1$ is equicontinuous. By the Arzelá–Ascoli theorem, $\mathcal{A}_1$ is completely continuous. By Krasnoselskiĭ's fixed-point theorem the $(k, \phi)$-Hilfer nonlocal multipoint fractional boundary value problem (16) has at least one solution on $[a,b]$. The proof is finished. □

**Theorem 3.** *Let* $\mathfrak{f} : [a,b] \times \mathbb{R} \to \mathbb{R}$ *be a continuous function. Assume that:*

*(H$_3$) there exist* $\chi : [0,\infty) \to (0,\infty)$ *which is continuous, nondecreasing function and a continuous positive function* $\sigma$ *such that*

$$|\mathfrak{f}(w,u)| \leq \sigma(w)\chi(|u|) \quad \text{for each} \quad (w,u) \in [a,b] \times \mathbb{R};$$

*(H$_4$) there exists a constant* $\mathfrak{K} > 0$ *such that*

$$\frac{\mathfrak{K}}{\chi(\mathfrak{K})\|\sigma\|\mathfrak{G}} > 1.$$

*Then the* $(k, \phi)$-*Hilfer nonlocal multipoint fractional boundary value problem (16) has at least one solution on* $[a,b]$.

**Proof.** In the first step we will show that the operator $\mathcal{A}$ maps bounded sets into bounded set in $C([a,b],\mathbb{R})$, where $\mathcal{A}$ is defined by (22). For $r > 0$, let $B_r = \{\vartheta \in C([a,b],\mathbb{R}) : \|\vartheta\| \leq r\}$. Then for $w \in [a,b]$ we have

$$
\begin{aligned}
&|(\mathcal{A}\vartheta)(w)| \\
&\leq \sup_{w \in [a,b]} \left\{ \frac{(\phi(w) - \phi(a))^{\frac{\theta_k}{k}-1}}{|\mathcal{H}|\Gamma_k(\theta_k)} \left[ \sum_{i=1}^{m} |\lambda_i|^k \mathfrak{J}^{\overline{\alpha};\phi}|\mathfrak{f}(\xi_i, \vartheta(\xi_i))| + {}^k\mathfrak{J}^{\overline{\alpha};\phi}|\mathfrak{f}(b, \vartheta(b))| \right] \right.\\
&\qquad\qquad \left. + {}^k\mathfrak{J}^{\overline{\alpha};\phi}|\mathfrak{f}(w, \vartheta(w))| \right\} \\
&\leq \left\{ \frac{(\phi(b) - \phi(a))^{\frac{\overline{\alpha}}{k}}}{\Gamma_k(\overline{\alpha} + k)} + \frac{(\phi(b) - \phi(a))^{\frac{\theta_k}{k}-1}}{|\mathcal{H}|\Gamma_k(\theta_k)} \left[ \sum_{i=1}^{m} |\lambda_i| \frac{(\phi(\xi_i) - \phi(a))^{\frac{\overline{\alpha}}{k}}}{\Gamma_k(\overline{\alpha} + k)} \right.\right.\\
&\qquad\qquad \left.\left. + \frac{(\phi(b) - \phi(a))^{\frac{\overline{\alpha}}{k}}}{\Gamma_k(\overline{\alpha} + k)} \right] \right\} \|\sigma\| \chi(\|\vartheta\|),
\end{aligned}
$$

and consequently,

$$
\|\mathcal{A}x\| \leq \chi(r)\|\sigma\|\mathfrak{G}.
$$

Now we will show that $\mathcal{A}$ maps bounded sets into equicontinuous sets of $C([a,b],\mathbb{R})$. Let $w_1, w_2 \in [a,b]$ with $w_1 < w_2$ and $\vartheta \in B_r$. Then we have

$$
\begin{aligned}
&|(\mathcal{A}\vartheta)(w_2) - (\mathcal{A}\vartheta)(w_1)| \\
&\leq \frac{1}{\Gamma_k(\overline{\alpha})} \left| \int_a^{w_1} \phi'(s)[(\phi(w_2) - \phi(s))^{\frac{\overline{\alpha}}{k}-1} - (\phi(w_1) - \phi(s))^{\frac{\overline{\alpha}}{k}-1}]\mathfrak{f}(s, \vartheta(s))ds \right.\\
&\qquad + \left. \int_{w_1}^{w_2} \phi'(s)(\phi(w_2) - \phi(s))^{\frac{\overline{\alpha}}{k}-1}\mathfrak{f}(s, \vartheta(s))ds \right| \\
&\qquad + \frac{(\phi(w_2) - \phi(a))^{\frac{\theta_k}{k}-1} - (\phi(w_1) - \phi(a))^{\frac{\theta_k}{k}-1}}{|\mathcal{H}|\Gamma_k(\theta_k)} \left[ \sum_{i=1}^{m} |\lambda_i|^k \mathfrak{J}^{\overline{\alpha};\phi}|\mathfrak{f}(\xi_i, \vartheta(\xi_i))| \right.\\
&\qquad \left. + {}^k\mathfrak{J}^{\overline{\alpha};\phi}|\mathfrak{f}(b, \vartheta(b))| \right] \\
&\leq \frac{\|\sigma\|\chi(r)}{\Gamma_k(\overline{\alpha} + k)} [2(\phi(w_2) - \phi(w_1))^{\frac{\overline{\alpha}}{k}} + |(\phi(w_2) - \phi(a))^{\frac{\overline{\alpha}}{k}} - (\phi(w_1) - \phi(a))^{\frac{\overline{\alpha}}{k}}|], \\
&\qquad + \frac{(\phi(w_2) - \phi(a))^{\frac{\theta_k}{k}-1} - (\phi(w_1) - \phi(a))^{\frac{\theta_k}{k}-1}}{|\mathcal{H}|\Gamma_k(\theta_k)} \left[ \sum_{i=1}^{m} |\lambda_i| \frac{(\phi(\xi_i) - \phi(a))^{\frac{\overline{\alpha}}{k}}}{\Gamma_k(\overline{\alpha} + k)} \right.\\
&\qquad \left. + \frac{(\phi(b) - \phi(a))^{\frac{\overline{\alpha}}{k}}}{\Gamma_k(\overline{\alpha} + k)} \right] \|\sigma\|\chi(r).
\end{aligned}
$$

As $w_2 - w_1 \to 0$ the right-hand side of the above inequality tends to zero independently of $\vartheta \in B_r$. Hence, the operator $\mathcal{A} : C([a,b],\mathbb{R}) \to C([a,b],\mathbb{R})$ is completely continuous, by the Arzelá–Ascoli theorem.

Finally we will show the boundedness of the set of all solutions to equations $\vartheta = \lambda\mathcal{A}\vartheta$ for $\lambda \in (0,1)$.

Let $\vartheta$ be a solution. Then, for $w \in [a,b]$, and working as in the first step, we have

$$
|\vartheta(w)| \leq \chi(\|\vartheta\|)\|\sigma\|\mathfrak{G},
$$

or

$$
\frac{\|\vartheta\|}{\chi(\|\vartheta\|)\|\sigma\|\mathfrak{G}} \leq 1.
$$

In view of $(H_4)$, there exists $\mathfrak{K}$ such that $\|\vartheta\| \neq \mathfrak{K}$. Let us set

$$U = \{\vartheta \in C([a,b],\mathbb{R}) : \|\vartheta\| < \mathfrak{K}\}.$$

We see that the operator $\mathcal{A} : \bar{U} \to C([a,b],\mathbb{R})$ is continuous and completely continuous. There is no $\vartheta \in \partial U$ such that $\vartheta = \lambda \mathcal{A} \vartheta$ for some $\lambda \in (0,1)$, from the choice of $U$. By the nonlinear alternative of Leray–Schauder type, we deduce that $\mathcal{A}$ has a fixed point $\vartheta \in \bar{U}$, which is a solution of the $(k,\phi)$-Hilfer nonlocal multipoint fractional boundary value problem (16). This completes the proof. $\square$

## 4. The Multivalued Problem

For a normed space $(\mathfrak{X}, \|\cdot\|)$, we define:
$\mathcal{P}_{cl}(\mathfrak{X}) = \{\mathfrak{R} \in \mathcal{P}(\mathfrak{X}) : \mathfrak{R} \text{ is closed}\}$, $\mathcal{P}_{cp}(\mathfrak{X}) = \{\mathfrak{R} \in \mathcal{P}(\mathfrak{X}) : \mathfrak{R} \text{ is compact}\}$, and $\mathcal{P}_{cp,c}(\mathfrak{X}) = \{\mathfrak{R} \in \mathcal{P}(\mathfrak{X}) : \mathfrak{R} \text{ is compact and convex}\}$.

For details of multivalued analysis we refer the reader to [27,28]. See also [7].

The set of selections of $\mathfrak{F}$, for each $\vartheta \in C([a,b],\mathbb{R})$, is defined by

$$S_{\mathfrak{F},\vartheta} := \{v \in L^1([a,b],\mathbb{R}) : v(w) \in \mathfrak{F}(w,\vartheta(w)) \text{ on } [a,b]\}.$$

**Definition 1.** *A function $\vartheta \in C([a,b],\mathbb{R})$ is said to be a solution of the $(k,\phi)$-Hilfer nonlocal multipoint fractional boundary value problem (17) if there exists a function $v \in L^1([a,b],\mathbb{R})$ with $v(w) \in \mathfrak{F}(w,\vartheta)$ for a.e. $w \in [a,b]$ such that $\vartheta$ satisfies the differential equation $^{k,H}\mathfrak{D}^{\alpha,\beta;\phi}\vartheta(w) = v(w)$ on $[a,b]$ and the boundary conditions $\vartheta(a) = 0$, $\vartheta(b) = \sum_{i=1}^m \lambda_i \vartheta(\xi_i)$.*

In the first existence result, which concern the case when $\mathfrak{F}$ has convex values, we apply nonlinear alternative of Leray–Schauder type [26] with the assumption that $\mathfrak{F}$ is $L^1$-Carathéodory, that is, (i) $w \to \mathfrak{F}(w,u)$ is measurable for each $u \in \mathbb{R}$; (ii) $u \to \mathfrak{F}(w,u)$ is upper semicontinuous for almost all $w \in [a,b]$ and (iii) for each $r > 0$, there exists a function $m_r \in L^1([a,b],\mathbb{R}^+)$ such that

$$\|\mathfrak{F}(w,u)\| = \sup\{|v| : v \in \mathfrak{F}(w,u)\} < m_r(w),$$

for each $u \in \mathbb{R}$ with $|u| \leq r$ and for almost every $w \in [a,b]$.

**Theorem 4.** *Assume that:*

$(G_1)$ *$\mathfrak{F} : [a,b] \times \mathbb{R} \to \mathcal{P}_{cp,c}(\mathbb{R})$ is $L^1$-Carathéodory;*
$(G_2)$ *there exists $z : [0,\infty) \to (0,\infty)$ a continuous nondecreasing function and a continuous positive function $q$ such that*

$$\|\mathfrak{F}(w,\vartheta)\|_{\mathcal{P}} := \sup\{|v| : v \in \mathfrak{F}(w,\vartheta)\} \leq q(w)z(\|\vartheta\|) \text{ for each } (w,\vartheta) \in [a,b] \times \mathbb{R};$$

$(G_3)$ *there exists a constant $\mathfrak{K} > 0$ such that*

$$\frac{\mathfrak{K}}{\|q\|z(\mathfrak{K})\mathfrak{G}} > 1.$$

*Then the $(k,\phi)$-Hilfer nonlocal multi-point fractional boundary value problem (17) has at least one solution on $[a,b]$.*

**Proof.** We define an operator $\mathcal{F} : C([a,b],\mathbb{R}) \longrightarrow \mathcal{P}(C([a,b],\mathbb{R}))$ by

$$\mathcal{F}(\vartheta) = \left\{ \begin{array}{l} h \in C([a,b],\mathbb{R}) : \\ h(w) = \left\{ \begin{array}{l} \dfrac{(\phi(w)-\phi(a))^{\frac{\theta_k}{k}-1}}{\mathcal{H}\Gamma_k(\theta_k)}\Big[\sum\limits_{i=1}^m \lambda_i\, ^k\mathfrak{J}^{\bar{\alpha};\phi}v(\xi_i,) - \,^k\mathfrak{J}^{\bar{\alpha};\phi}v(b)\Big] \\ +\,^k\mathfrak{J}^{\bar{\alpha};\phi}v(w),\ w \in [a,b] \end{array} \right. \end{array} \right\}$$

and $v \in S_{\mathfrak{F},\vartheta}$. It is obvious that the solutions of the $(k,\phi)$-Hilfer nonlocal multipoint fractional boundary value problem (17) are the fixed points of $\mathcal{F}$.

We will give the proof in several steps.

*Step 1. For each $\vartheta \in C([a,b],\mathbb{R})$, the operator $\mathcal{F}(\vartheta)$ is convex.*

We omit the proof, because it is obvious, since $\mathfrak{F}$ has convex values and thus $S_{F,\vartheta}$ is convex.

*Step 2. $\mathcal{F}$ maps the bounded sets into bounded sets in $C([a,b],\mathbb{R})$.*

Let $B_r = \{\vartheta \in C([a,b],\mathbb{R}) : \|\vartheta\| \le r\}, r > 0$. Then, for each $h \in \mathcal{F}(\vartheta), \vartheta \in B_r$, there exists $v \in S_{\mathfrak{F},x}$ such that

$$h(w) = \frac{(\phi(w) - \phi(a))^{\frac{\theta_k}{k}-1}}{\mathcal{H}\Gamma_k(\theta_k)} \left[ \sum_{i=1}^{m} \lambda_i {}^k\mathfrak{J}^{\overline{\alpha};\phi}v(\xi_i,) - {}^k\mathfrak{J}^{\overline{\alpha};\phi}v(b) \right] + {}^k\mathfrak{J}^{\overline{\alpha};\phi}v(w).$$

Then, for $w \in [a,b]$, we have

$$
\begin{aligned}
|h(w)| &\le \sup_{w \in [a,b]} \left\{ \frac{(\phi(w) - \phi(a))^{\frac{\theta_k}{k}-1}}{|\mathcal{H}|\Gamma_k(\theta_k)} \left[ \sum_{i=1}^{m} |\lambda_i| {}^k\mathfrak{J}^{\overline{\alpha};\phi}|v(\xi_i)| + {}^k\mathfrak{J}^{\overline{\alpha};\phi}|v(b)| \right] \right. \\
&\quad \left. + {}^k\mathfrak{J}^{\overline{\alpha};\phi}|v(w)| \right\} \\
&\le \left\{ \frac{(\phi(b) - \phi(a))^{\frac{\overline{\alpha}}{k}}}{\Gamma_k(\overline{\alpha}+k)} + \frac{(\phi(b) - \phi(a))^{\frac{\theta_k}{k}-1}}{|\mathcal{H}|\Gamma_k(\theta_k)} \left[ \sum_{i=1}^{m} |\lambda_i| \frac{(\phi(\xi_i) - \phi(a))^{\frac{\overline{\alpha}}{k}}}{\Gamma_k(\overline{\alpha}+k)} \right. \right. \\
&\quad \left. \left. + \frac{(\phi(b) - \phi(a))^{\frac{\overline{\alpha}}{k}}}{\Gamma_k(\overline{\alpha}+k)} \right] \right\} \|q\| z(\|\vartheta\|),
\end{aligned}
$$

and consequently,

$$\|h\| \le z(r)\|q\|\mathfrak{G}.$$

*Step 3. $\mathcal{F}$ maps bounded sets into equicontinuous sets of $C([a,b],\mathbb{R})$.*

Let $w_1, w_2 \in [a,b]$ with $w_1 < w_2$ and $\vartheta \in B_r$. Then, for each $h \in \mathcal{F}(\vartheta)$, we obtain

$$
\begin{aligned}
&|h(w_2) - h(w_1)| \\
&\le \frac{1}{\Gamma_k(\alpha)} \left| \int_a^{w_1} \phi'(s)[(\phi(w_2) - \phi(s))^{\frac{\overline{\alpha}}{k}-1} - (\phi(w_1) - \phi(s))^{\frac{\overline{\alpha}}{k}-1}]v(s)ds \right. \\
&\quad \left. + \int_{w_1}^{w_2} \phi'(s)(\phi(w_2) - \phi(s))^{\frac{\overline{\alpha}}{k}-1}v(s)ds \right| \\
&\quad + \frac{(\phi(w_2) - \phi(a))^{\frac{\theta_k}{k}-1} - (\phi(w_1) - \phi(a))^{\frac{\theta_k}{k}-1}}{|\mathcal{H}|\Gamma_k(\theta_k)} \left[ \sum_{i=1}^{m} |\lambda_i| {}^k\mathfrak{J}^{\overline{\alpha};\phi}|v(\xi_i)| \right. \\
&\quad \left. + {}^k\mathfrak{J}^{\overline{\alpha};\phi}|v(b)| \right] \\
&\le \frac{\|q\|z(r)}{\Gamma_k(\overline{\alpha}+k)} [2(\phi(w_2) - \phi(w_1))^{\frac{\overline{\alpha}}{k}} + |(\phi(w_2) - \phi(a))^{\frac{\overline{\alpha}}{k}} - (\phi(w_1) - \phi(a))^{\frac{\overline{\alpha}}{k}}|], \\
&\quad + \frac{(\phi(w_2) - \phi(a))^{\frac{\theta_k}{k}-1} - (\phi(w_1) - \phi(a))^{\frac{\theta_k}{k}-1}}{|\mathcal{H}|\Gamma_k(\theta_k)} \left[ \sum_{i=1}^{m} |\lambda_i| \frac{(\phi(\xi_i) - \phi(a))^{\frac{\overline{\alpha}}{k}}}{\Gamma_k(\overline{\alpha}+k)} \right. \\
&\quad \left. + \frac{(\phi(b) - \phi(a))^{\frac{\overline{\alpha}}{k}}}{\Gamma_k(\overline{\alpha}+k)} \right] \|q\|z(r).
\end{aligned}
$$

Hence, independently of $\vartheta \in B_r$ we have $|h(w_2) - h(w_1)| \to 0$ as $w_2 - w_1 \to 0$. By the Arzelá–Ascoli theorem that $\mathcal{F} : C([a,b], \mathbb{R}) \to \mathcal{P}(C([a,b], \mathbb{R}))$ is completely continuous.

By virtue of the Proposition 1.2 of [24], it is enough to prove that the $\mathcal{F}$ has a closed graph, which will imply that $\mathcal{F}$ is upper semicontinuous multivalued mapping.

*Step 4. $\mathcal{F}$ has a closed graph.*

Let $\vartheta_n \to \vartheta_*, h_n \in \mathcal{F}(\vartheta_n)$ and $h_n \to h_*$. Then we need to show that $h_* \in \mathcal{F}(\vartheta_*)$. Associated with $h_n \in \mathcal{F}(\vartheta_n)$, there exists $v_n \in S_{\mathfrak{F}, \vartheta_n}$ such that for each $w \in [a, b]$,

$$h_n(w) = \frac{(\phi(w) - \phi(a))^{\frac{\theta_k}{k} - 1}}{\mathcal{H}\Gamma_k(\theta_k)} \left[ \sum_{i=1}^{m} \lambda_i {}^k \mathfrak{J}^{\bar{\alpha}; \phi} v_n(\xi_i,) - {}^k \mathfrak{J}^{\bar{\alpha}; \phi} v_n(b) \right] + {}^k \mathfrak{J}^{\bar{\alpha}; \phi} v_n(w).$$

Thus it suffices to show that there exists $v_* \in S_{\mathfrak{F}, \vartheta_*}$ such that for each $w \in [a, b]$,

$$h_*(w) = \frac{(\phi(w) - \phi(a))^{\frac{\theta_k}{k} - 1}}{\mathcal{H}\Gamma_k(\theta_k)} \left[ \sum_{i=1}^{m} \lambda_i {}^k \mathfrak{J}^{\bar{\alpha}; \phi} v_*(\xi_i) - {}^k \mathfrak{J}^{\bar{\alpha}; \phi} v_*(b) \right] + {}^k \mathfrak{J}^{\bar{\alpha}; \phi} v_*(w).$$

Let us consider the linear operator $\Theta : L^1([a,b], \mathbb{R}) \to C([a,b], \mathbb{R})$ given by

$$v \mapsto \Theta(v)(w) \frac{(\phi(w) - \phi(a))^{\frac{\theta_k}{k} - 1}}{\mathcal{H}\Gamma_k(\theta_k)} \left[ \sum_{i=1}^{m} \lambda_i {}^k \mathfrak{J}^{\bar{\alpha}; \phi} v(\xi_i) - {}^k \mathfrak{J}^{\bar{\alpha}; \phi} v(b) \right] + {}^k \mathfrak{J}^{\bar{\alpha}; \phi} v(w).$$

Observe that $\|h_n - h_*\| \to 0$, as $n \to \infty$. Therefore, it follows by a Lazota–Opial result [29], that $\Theta \circ S_{\mathfrak{F}}$ is a closed-graph operator. Further, we have $h_n(w) \in \Theta(S_{\mathfrak{F}, \vartheta_n})$. Since $\vartheta_n \to \vartheta_*$, we have

$$h_*(w) = \frac{(\phi(w) - \phi(a))^{\frac{\theta_k}{k} - 1}}{\mathcal{H}\Gamma_k(\theta_k)} \left[ \sum_{i=1}^{m} \lambda_i {}^k \mathfrak{J}^{\bar{\alpha}; \phi} v_*(\xi_i) - {}^k \mathfrak{J}^{\bar{\alpha}; \phi} v_*(b) \right] + {}^k \mathfrak{J}^{\bar{\alpha}; \phi} v_*(w),$$

for some $v_* \in S_{\mathfrak{F}, \vartheta_*}$.

*Step 5. There exists an open set $\mathcal{U} \subseteq C([a,b], \mathbb{R})$ with $\vartheta \notin \nu \mathcal{F}(\vartheta)$ for any $\nu \in (0, 1)$ and all $\vartheta \in \partial \mathcal{U}$.*

Let $\nu \in (0, 1)$ and $\vartheta \in \nu \mathcal{F}(\vartheta)$. Then there exists $v \in L^1([a,b], \mathbb{R})$ with $v \in S_{\mathfrak{F}, \vartheta}$ such that, for $w \in [a, b]$, we have

$$\vartheta(w) = \nu \frac{(\phi(w) - \phi(a))^{\frac{\theta_k}{k} - 1}}{\mathcal{H}\Gamma_k(\theta_k)} \left[ \sum_{i=1}^{m} \lambda_i {}^k \mathfrak{J}^{\bar{\alpha}; \phi} v(\xi_i) - {}^k \mathfrak{J}^{\bar{\alpha}; \phi} v(b) \right] + \nu {}^k \mathfrak{J}^{\bar{\alpha}; \phi} v(w).$$

Working as in second step, we have

$$|\vartheta(w)| \le \|p\| \omega(\|\vartheta\|) \mathfrak{G}.$$

Consequently

$$\|\vartheta\| \le \|p\| \omega(\|\vartheta\|) \mathfrak{G},$$

or

$$\frac{\|\vartheta\|}{\|q\| z(\|\vartheta\|) \mathfrak{G}} \le 1.$$

In view of $(H_3)$, there exists $\mathfrak{K}$ such that $\|\vartheta\| \neq \mathfrak{K}$. Let us set

$$\mathcal{U} = \{\vartheta \in C([a,b], \mathbb{R}) : \|\vartheta\| < \mathfrak{K}\}.$$

The operator $\mathcal{F} : \overline{\mathcal{U}} \to \mathcal{P}(C([a,b],\mathbb{R}))$ is a compact multivalued map, upper semicontinuous with convex closed values. There is no $\vartheta \in \partial \mathcal{U}$ such that $\vartheta \in \nu \mathcal{F}(\vartheta)$ for some $\nu \in (0,1)$, from the choice of $\mathcal{U}$.

By the nonlinear alternative of Leray–Schauder type $\mathcal{F}$ has a fixed point $\vartheta \in \overline{\mathcal{U}}$ which is a solution of the $(k,\phi)$-Hilfer nonlocal multi-point fractional boundary value problem (17). This ends the proof. □

In our second result, the existence of solutions for the $(k,\phi)$-Hilfer nonlocal multipoint fractional boundary value problem (17) is showed when $F$ is not necessarily nonconvex valued by using a fixed-point theorem for multivalued contractive maps due to Covitz and Nadler [30].

**Theorem 5.** *Assume that the following conditions hold:*

$(A_1)\, \mathfrak{F} : [a,b] \times \mathbb{R} \to \mathcal{P}_{cp}(\mathbb{R})$ *is such that* $\mathfrak{f}(\cdot, \vartheta) : [a,b] \to \mathcal{P}_{cp}(\mathbb{R})$ *is measurable for each* $\vartheta \in \mathbb{R}$.
$(A_2)\, H_d(\mathfrak{F}(w,\vartheta), \mathfrak{F}(w,\bar{\vartheta})) \leq m(w)|\vartheta - \bar{\vartheta}|$ *for almost all* $w \in [a,b]$ *and* $\vartheta, \bar{\vartheta} \in \mathbb{R}$ *with* $m \in C([a,b],\mathbb{R}^+)$ *and* $d(0,\mathfrak{f}(w,0)) \leq m(w)$ *for almost all* $w \in [a,b]$.

*Then the* $(k,\phi)$-*Hilfer nonlocal multipoint fractional boundary value problem* (17) *has at least one solution on* $[a,b]$ *if*

$$\delta := \mathfrak{G}\|m\| < 1. \tag{27}$$

**Proof.** By the assumption $(A_1)$, the set $S_{\mathfrak{F},\vartheta}$ is nonempty for each $\vartheta \in C([a,b],\mathbb{R})$. Hence $\mathfrak{F}$ has a measurable selection (see Theorem III.6 [31]). We show that $\mathcal{F}(\vartheta) \in \mathcal{P}_{cl}(C([a,b],\mathbb{R}))$ for each $\vartheta \in C([a,b],\mathbb{R})$. Let $\{u_n\}_{n \geq 0} \in \mathcal{F}(\vartheta)$ be such that $u_n \to u\ (n \to \infty)$ in $C([a,b],\mathbb{R})$. Then $u \in C([a,b],\mathbb{R})$ and there exists $v_n \in S_{\mathfrak{F},\vartheta_n}$ such that, for each $w \in [a,b]$,

$$u_n(w) = \frac{(\phi(w) - \phi(a))^{\frac{\theta_k}{k}-1}}{\mathcal{H}\Gamma_k(\theta_k)} \left[ \sum_{i=1}^{m} \lambda_i\, {}^k\mathcal{J}^{\bar{\alpha};\phi} v_n(\xi_i) - {}^k\mathcal{J}^{\bar{\alpha};\phi} v_n(b) \right] + {}^k\mathcal{J}^{\bar{\alpha};\phi} v_n(w).$$

As $\mathfrak{F}$ has compact values, we pass onto a subsequence (if necessary) to obtain that $v_n$ converges to $v$ in $L^1([a,b],\mathbb{R})$. Thus, $v \in S_{\mathfrak{F},\vartheta}$ and for each $w \in [a,b]$, we have

$$u_n(w) \to u(w) = \frac{(\phi(w) - \phi(a))^{\frac{\theta_k}{k}-1}}{\mathcal{H}\Gamma_k(\theta_k)} \left[ \sum_{i=1}^{m} \lambda_i\, {}^k\mathcal{J}^{\bar{\alpha};\phi} v_n(\xi_i) - {}^k\mathcal{J}^{\bar{\alpha}\phi} v_n(b) \right] + {}^k\mathcal{J}^{\bar{\alpha};\phi} v_n(w).$$

Hence, $u \in \mathcal{F}(\vartheta)$.

Next we show that

$$H_d(\mathcal{F}(\vartheta), \mathcal{F}(\bar{\vartheta})) \leq \delta\|\vartheta - \bar{\vartheta}\|, \quad \delta < 1, \quad \text{for each} \ \ \vartheta, \bar{\vartheta} \in C^2([a,b],\mathbb{R}).$$

Let $\vartheta, \bar{\vartheta} \in C^2([a,b],\mathbb{R})$ and $h_1 \in \mathcal{F}(x)$. Then there exists $v_1(w) \in \mathfrak{F}(w,\vartheta(w))$ such that, for each $w \in [a,b]$,

$$h_1(w) = \frac{(\phi(w) - \phi(a))^{\frac{\theta_k}{k}-1}}{\mathcal{H}\Gamma_k(\theta_k)} \left[ \sum_{i=1}^{m} \lambda_i\, {}^k\mathcal{J}^{\bar{\alpha};\phi} v_1(\xi_i) - {}^k\mathcal{J}^{\bar{\alpha};\phi} v_1(b) \right] + {}^k\mathcal{J}^{\bar{\alpha};\phi} v_1(w).$$

By $(A_2)$, we have

$$H_d(\mathfrak{F}(w,\vartheta), \mathfrak{F}(w,\bar{\vartheta})) \leq m(w)|\vartheta(w) - \bar{\vartheta}(w)|.$$

So, there exists $\omega \in \mathfrak{f}(w,\bar{x}(w))$ such that

$$|v_1(w) - \omega| \leq m(w)|\vartheta(w) - \bar{\vartheta}(w)|, \ \ w \in [a,b].$$

Define $U : [a,b] \to \mathcal{P}(\mathbb{R})$ by

$$U(w) = \{w \in \mathbb{R} : |v_1(w) - \omega| \le m(w)|\vartheta(w) - \bar{\vartheta}(w)|\}.$$

Since the multivalued operator $U(w) \cap \mathfrak{F}(w, \bar{\vartheta}(w))$ is measurable (Proposition III.4 [31]), there exists a function $v_2(w)$ which is a measurable selection for $U$. So $v_2(w) \in \mathfrak{F}(w, \bar{\vartheta}(w))$ and for each $w \in [a,b]$, we have $|v_1(w) - v_2(w)| \le m(w)|\vartheta(w) - \bar{\vartheta}(w)|$.

For each $w \in [a,b]$, let us define

$$h_2(w) = \frac{(\phi(w) - \phi(a))^{\frac{\theta_k}{k} - 1}}{\mathcal{H}\Gamma_k(\theta_k)} \left[ \sum_{i=1}^{m} \lambda_i{}^k \mathcal{J}^{\bar{\alpha};\phi} v_2(\xi_i) - {}^k \mathcal{J}^{\bar{\alpha};\phi} v_2(b) \right] + {}^k \mathcal{J}^{\bar{\alpha};\phi} v_2(w).$$

Thus,

$$
\begin{aligned}
&|h_1(w) - h_2(w)| \\
\le\ & \frac{(\phi(w) - \phi(a))^{\frac{\theta_k}{k} - 1}}{\mathcal{H}\Gamma_k(\theta_k)} \left[ \sum_{i=1}^{m} \lambda_i{}^k \mathcal{J}^{\bar{\alpha};\phi}(|v_1(s) - v_2(s)|)(\xi_i) + {}^k \mathcal{J}^{\bar{\alpha};\phi}(|v_1(s) - v_2(s)|)(b) \right] \\
& + {}^k \mathcal{J}^{\bar{\alpha};\phi}(|v_1(s) - v_2(s)|)(w) \\
\le\ & \left\{ \frac{(\phi(b) - \phi(a))^{\frac{\bar{\alpha}}{k}}}{\Gamma_k(\bar{\alpha} + k)} + \frac{(\phi(b) - \phi(a))^{\frac{\theta_k}{k} - 1}}{|\mathcal{H}|\Gamma_k(\theta_k)} \left[ \sum_{i=1}^{m} |\lambda_i| \frac{(\phi(\xi_i) - \phi(a))^{\frac{\bar{\alpha}}{k}}}{\Gamma_k(\bar{\alpha} + k)} \right. \right. \\
& \left. \left. + \frac{(\phi(b) - \phi(a))^{\frac{\bar{\alpha}}{k}}}{\Gamma_k(\bar{\alpha} + k)} \right] \right\} \|m\| \|\vartheta - \bar{\vartheta}\| \\
=\ & \mathfrak{G}\|m\| \|\vartheta - \bar{\vartheta}\|.
\end{aligned}
$$

Hence

$$\|h_1 - h_2\| \le \mathfrak{G}\|m\| \|\vartheta - \bar{\vartheta}\|.$$

Analogously, interchanging the roles of $x$ and $\bar{x}$, we obtain

$$H_d(\mathcal{F}(\vartheta), \mathcal{F}(\bar{\vartheta})) \le \mathfrak{G}\|m\| \|\vartheta - \bar{v}\|.$$

So $\mathcal{F}$ is a contraction and by Covitz and Nadler theorem $\mathcal{F}$ has a fixed point $\vartheta$ which is a solution of the $(k, \phi)$-Hilfer nonlocal multipoint fractional boundary value problem (17). This completes the proof. $\square$

## 5. Examples

Now, we present some examples to show the applicability of our results.

**Example 1.** *Consider the following multipoint boundary value problems for $(k, \phi)$-Hilfer fractional derivative of the form*

$$
\begin{cases}
\frac{1}{6}, H \mathfrak{D}^{\frac{3}{2}, \frac{4}{5}; w^7 e^{-2w}} \vartheta(w) = \mathfrak{f}(w, \vartheta(w)), & \frac{1}{5} < w < \frac{8}{5}, \\[2mm]
\vartheta\left(\frac{1}{5}\right) = 0, \qquad \vartheta\left(\frac{8}{5}\right) = \frac{1}{11}\vartheta\left(\frac{2}{5}\right) + \frac{3}{22}\vartheta\left(\frac{3}{5}\right) \\[2mm]
\qquad\qquad\qquad + \frac{5}{33}\vartheta\left(\frac{4}{5}\right) + \frac{7}{44}\vartheta\left(\frac{6}{5}\right) + \frac{9}{55}\vartheta\left(\frac{7}{5}\right).
\end{cases}
\tag{28}
$$

Here $\bar{\alpha} = 3/2$, $\beta = 4/5$, $\phi(w) = w^7 e^{-2w}$, $k = 1/6$, $a = 1/5$, $b = 8/5$, $m = 5$, $\lambda_1 = 1/11$, $\lambda_2 = 3/22$, $\lambda_3 = 5/33$, $\lambda_4 = 7/44$, $\lambda_5 = 9/55$, $\xi_1 = 2/5$, $\xi_2 = 3/5$, $\xi_3 = 4/5$, $\xi_4 = 6/5$, $\xi_5 = 7/5$. By direct computation, we get $\theta_{\frac{1}{6}} = 17/30$, $\Gamma_{\frac{1}{6}}(\theta_{\frac{1}{6}}) \approx 0.04044166691$, $\mathcal{H} \approx 29.03126784$, $\mathfrak{G} \approx 128.5303681$, $\mathfrak{G}_1 \approx 66.09288339$.

(i) Let a nonlinear unbounded $\mathfrak{f}(w, \vartheta)$ be given by

$$\mathfrak{f}(w, \vartheta) = \frac{e^{-(5w-1)^2}}{40(5w+6)}\left(\frac{\vartheta^2 + |\vartheta|}{1+|\vartheta|}\right) + \frac{1}{3}w + \frac{1}{2}. \tag{29}$$

Then we can show that,

$$|\mathfrak{f}(w, \vartheta_1) - \mathfrak{f}(w, \vartheta_2)| \le \frac{1}{140}|\vartheta_1 - \vartheta_2|,$$

for $w \in [1/5, 8/5]$ and $\vartheta_1, \vartheta_2 \in \mathbb{R}$. Therefore, for $\mathfrak{L} = 1/140$, we have $\mathfrak{L}\mathfrak{G} \approx 0.9180740579 < 1$. Thus by Theorem 1 the multipoint boundary value problem for $(k, \phi)$-Hilfer fractional derivative (28) with (29) has a unique solution on the interval $[1/5, 8/5]$.

(ii) Let a nonlinear bounded $\mathfrak{f}(w, \vartheta)$ be defined as

$$\mathfrak{f}(w, \vartheta) = \frac{e^{-(5w-1)^2}}{10(5w+6)}\left(\frac{|\vartheta|}{1+|\vartheta|}\right) + \frac{1}{3}w + \frac{1}{2}. \tag{30}$$

Now, we observe that

$$|\mathfrak{f}(w, \vartheta)| \le \frac{e^{-(5w-1)^2}}{10(5w+6)} + \frac{1}{3}w + \frac{1}{2} := \varpi(w),$$

which is bounded by the known function $\varpi(w)$, $w \in [1/5, 8/5]$. In addition, $\mathfrak{f}$ satisfies the Lipschitz condition $(H_1)$ with Lipschitz constant $\mathfrak{L} = 1/70$. But we can not conclude the uniqueness result, because Theorem 1 can not be applied since $\mathfrak{L}\mathfrak{G} \approx 1.836148116 > 1$. However, since $\mathfrak{L}\mathfrak{G}_1 \approx 0.9441840484 < 1$, we deduce that the boundary value problem (28), with $\mathfrak{f}$ given by (30), has at least one solution on $[1/5, 8/5]$ by Theorem 2.

(iii) Let now a nonlinear $\mathfrak{f}(w, \vartheta)$ be presented by

$$\mathfrak{f}(w, \vartheta) = \frac{1}{2(5w+7)}\left(\frac{\vartheta^{182}}{15(1+\vartheta^{180})} + \frac{1}{18}\right). \tag{31}$$

Note that the nonlinear function can be bounded by quadratic term as

$$|\mathfrak{f}(w, \vartheta)| \le \frac{1}{2(5w+7)}\left(\frac{1}{15}\vartheta^2 + \frac{1}{18}\right).$$

By setting $\sigma(w) = 1/(2(5w+7))$ and $\chi(u) = (1/15)u^2 + (1/18)$, we have $\|\sigma\| = 1/16$ and, then, there exists $\mathfrak{K} \in (0.7378396700, 1.129423324)$ satisfying condition $(H_4)$ in Theorem 3. By application of Theorem 3, we conclude that the multipoint boundary value problem via $(k, \phi)$-Hilfer fractional calculus (28), with $\mathfrak{f}$ given by (30), has at least one solution on $[1/5, 8/5]$.

(iv) Let the first equation of (28) be replaced by

$$^{\frac{1}{6}, H}\mathfrak{D}^{\frac{3}{2}, \frac{4}{5}; w^7 e^{-2w}}\vartheta(w) \in \mathfrak{F}(w, \vartheta(w)), \quad \frac{1}{5} < w < \frac{8}{5}, \tag{32}$$

where

$$\mathfrak{F}(w, \vartheta) = \left[0, \frac{1}{20(5w+12)}\left(\frac{|\vartheta|}{1+|\vartheta|} + \sin\vartheta + 1\right)\right].$$

Now, we see that $\mathfrak{F}(w, \vartheta)$ is a measurable set. In addition, we have

$$H_d\big(\mathfrak{F}(w, \vartheta), \mathfrak{F}(w, \overline{\vartheta})\big) \le \frac{1}{10(5w+12)}|\vartheta - \overline{\vartheta}|.$$

We set $m(w) = 1/(10(5w+12))$. Therefore, we can check that $d(0, \mathfrak{F}(w, 0)) \le 1/(20(5w+12)) \le 1/(10(5w+12)) = m(w)$ for almost all $w \in [1/5, 8/5]$. As $\delta =$

$\mathfrak{G}\|m\| \approx 0.9886951392 < 1$, we get that $(k, \phi)$-Hilfer fractional inclusion (32) with boundary conditions given in (28), has at least one solution on $[1/5, 8/5]$.

## 6. Conclusions

In the present research, we have investigated fractional boundary value problems consisting of $(k, \phi)$-Hilfer fractional differential equations and inclusions, supplemented by nonlocal multipoint boundary conditions. First we considered the single valued case. After transforming the given problem into a fixed-point problem, we applied the Banach contraction-mapping principle, the Krasnoselskiĭ fixed-point theorem and the Leray–Schauder nonlinear alternative and established existence and uniqueness results. After that, we studied the multivalued case. We considered both cases, convex-valued and nonconvex-valued multivalued maps. In the first case, we established an existence result via a Leray–Schauder nonlinear alternative for multivalued maps, while in the second case the Covitz–Nadler fixed-point theorem for contractive multivalued maps was applied. Numerical examples illustrating the theoretical results are also presented. The used methods are standard, but their configuration in $(k, \phi)$-Hilfer nonlocal multipoint fractional boundary value problems is new. To the best of our knowledge, our results in this paper are the only concerning boundary value problems involving $(k, \phi)$-Hilfer fractional differential equations and inclusions of order in $(1, 2]$. Hence our results will enrich the literature on this new research area.

**Author Contributions:** Conceptualization, J.T., A.S. and S.K.N.; methodology, J.T., A.S. and S.K.N.; validation, J.T., A.S. and S.K.N.; formal analysis, J.T., A.S. and S.K.N.; writing—original draft preparation, J.T., A.S. and S.K.N.; funding acquisition, J.T. All authors have read and agreed to the published version of the manuscript.

**Funding:** This research was funded by National Science, Research and Innovation Fund (NSRF), and King Mongkut's University of Technology North Bangkok with Contract no. KMUTNB-FF-65-36.

**Institutional Review Board Statement:** Not applicable.

**Informed Consent Statement:** Not applicable.

**Data Availability Statement:** Not applicable.

**Conflicts of Interest:** The authors declare no conflict of interest.

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
