# Peer review of "Multi-Point Boundary Value Problems for (k, ϕ)-Hilfer Fractional Differential Equations and Inclusions"

_axioms, doi:10.3390/axioms11030110_

Round 1
Reviewer 1 Report
Please see the attachment: REPORT ON axioms-1609376.doc

Author Response
Response to Reviewer- 1

Reviewer 2 Report
The paper studied boundary value problems for fractional differential
equations and inclusions involving (k, phi)-Hilfer fractional derivative of order in (1, 2]. In the single valued case the existence and uniqueness results are established. In the multi valued case both convex or non-convex values for the right-hand side are considered. The (k, phi)-Hilfer fractional derivatives are of great interests because they extend many types of fractional derivatives in the literature. This paper seems to be the first one to study the existence and uniqueness of the solutions of the boundary value problems. Subject to some mild constraints, existence and uniqueness results are obtained. The proofs are rigorous and well organized. A concrete numerical example is given as the application of main findings. The authors indeed solved a new problem in the open area with useful applications. Therefore I recommend acceptance of the paper subject to minor review.
Below is the list of typo that I have found. There exist more such typo in the paper, please double check them before submitting the revised version.
(1) Page 1, Paragraph 1, Line 3: change "word" to "world".
(2) In Equations (2) and (3): change n-a to n-alpha.
(3) Page 2, Line 2: change "were" to "was".
(4) Page 2, Line 2 after Equation (8): change "were" to "was".
(5) Page 3, Items 2, 3, 4: provide references for each particular case of (k, phi)-Hilfer fractional derivative.
(6) In the paper, some "Leray-Schauder" is written "Laray-Schauder". Please correct them.
(7) Page 6, Line 85: should assume v\in B_r in the last inequality.
(8) Page 9, Line 2: Add dot to end the phrase.
(9) Page 9, in (H_3): add space between "\sigma" and "such".
(10) Page 10, in the last inequality between Line 121 and 122, remove ",".
(11) Page 11, Line 145, should w\in[a,b] inside the parenthesis of the right-hand side of F(v)?
(12) Page 11, Lines 148 and 149, the two sentences need to be rephrased.
(13) Page 14, add "," after the inequality between Line 197 and 198.
Author Response
Response to Reviewer-2

Reviewer 3 Report
This paper was motivated by a nonlinear initial value problem involving (k, φ)-Hilfer fractional derivative studied in [16]. Thus the authors initiate the study of nonlocal multi-point boundary conditions involving (k, φ)-Hilfer fractional derivative operator of order a from (1, 2] and parameter b from [0,1] (see the problem (16) from the paper). The given problem is transformed into a fixed point problem and by applying Banach contraction mapping principle, Krasnoselski˘i fixed point theorem and the Leray–Schauder nonlinear alternative is established the existence and uniqueness of the solution. Next they also studied a similar multivalued problem when the right-hand side is convex or non-convex valued. An existence result is obtained by Leray-Schauder nonlinear alternative for multivalued maps in the first case and by the Covitz-Nadler fixed point theorem for contractive multivalued maps in the second case. The applicability of the theoretical results is illustrated by numerical examples.
The paper is well-written and provide sufficient background. The proofs of the theoretical results are clear and the numerical examples are well presented.
As minor remarks notice that: in p.2, line 2 and line 6 below, must replace “where” by “was” and I did not found the references [25], [26] quoted in the text.
Author Response
Response to Reviewer-3
